# Fungal Endophytes Enhance Wheat and Tomato Drought Tolerance in Terms of Plant Growth and Biochemical Parameters

**DOI:** 10.3390/jof9030384

**Published:** 2023-03-21

**Authors:** Victoria Miranda, Gloria Andrea Silva-Castro, Juan Manuel Ruiz-Lozano, Sebastian Fracchia, Inmaculada García-Romera

**Affiliations:** 1Centro Regional de Investigaciones Científicas y Transferencia Tecnológica de La Rioja (CRILAR-CONICET, Provincia de La Rioja, UNLAR, SEGEMAR, UNCa), Entre Ríos y Mendoza s/n, Anillaco La Rioja 5301, Argentina; 2Department of Soil Microbiology and Symbiotic Systems, Estación Experimental del Zaidín, Consejo Superior de Investigaciones Científicas, Prof. Albareda 1 Apdo. 419, E-18008 Granada, Spain; 3The Mycology Laboratory, PROPLAME-PRHIDEB-CONICET, Department of Biodiversity and Experimental Biology, Faculty of Exact and Natural Sciences, University of Buenos Aires, Buenos Aires 1053, Argentina

**Keywords:** fungal endophytes, *Zopfiella* strains, drought tolerance, oxidative damage

## Abstract

Drought is a major threat to plant growth in many parts of the world. During periods of drought, multiple aspects of plant physiology are negatively affected. For instance, water shortages induce osmotic imbalance, inhibit photosynthesis, decrease nutrient uptake, and increases the production of reactive oxygen species (ROS). In this context, it is necessary to develop sustainable strategies for crops that would help mitigate these conditions. In previous studies, endophytic *Zopfiella erostrata* strains were found to extensively colonize plant roots, forming a profuse melanized mycelium in the rhizosphere, which could be involved in improving water uptake and nutrient mineralization in plants. The aim of this study is to evaluate the effect of different strains of *Z. erostrata* on stress mitigation in wheat and tomato plants grown under water deficit conditions. General plant growth variables, as well as physiological and biochemical parameters, related to oxidative status were determined. Our data demonstrate that inoculation with both *Zopfiella* strains had a very significant effect on plant growth, even under water deficit conditions. However, we observed an even more pronounced impact, depending on the plant and strain involved, suggesting a certain degree of plant/strain compatibility. The biochemical aspects, the accumulation of proline, the oxidative damage to lipids, and the activity of antioxidant enzymes varied considerably depending on the endophyte and the plant evaluated.

## 1. Introduction

In agricultural environments, most crops are exposed to a diverse range of biotic and abiotic stresses, which profoundly affect their development and productivity, due to the inhibition of essential physiological and metabolic processes [1,2,3]. Drought represents a critical plant growth constraint in many parts of the world [4,5]. During drought stress, multiple aspects of plant physiology are negatively affected [6]. For instance, water deficit induces osmotic imbalance that produces turgor loss, and the inhibition of photosynthetic capacity also diminishes nutrient uptake and transport to the aerial part, leading to hormonal and nutritional alterations in the plant [7,8]. In addition, drought stress increases the production of reactive oxygen species (ROS), which causes oxidative damage to biomolecules [9,10]. This increase in ROS can alter membrane permeability, damages proteins, DNA, and active lipid peroxidation products, leading ultimately to cell death in plant tissues [11,12].

55% of the total area of Argentina is made up of drylands (arid, semi-arid, and dry sub-humid ecosystems), with temperature extremes and a variable moisture regime restricted mainly to the summer months [13]. These regions are generally characterized by poorly developed high drainage soils with low water-holding capacity and organic matter content [14]. Due to their physicochemical features associated with the adverse climatic conditions, these soils are highly susceptible to degradation processes, which are becoming more acute every year [15,16,17]. In this context, it is necessary to develop sustainable strategies to help mitigate drought stress conditions affecting crops. Among these strategies of sustainable management, numerous researchers have suggested that fungal endophytes have the capacity to protect host plants against water stress [18,19,20]. The searching for endophytes with the potential to alleviate the damage caused by drought stress in plants could be a promising avenue of research aimed at improving traditional agricultural practices.

Dry ecosystem plants harbor a wide variety of fungal endophytes, which grow in the root tissues of fungi without producing disease symptoms [21,22,23]. These endophytes are major colonizers, especially in extreme environments, where they promote tolerance to drought and stimulate plant growth under the stressful conditions outlined above [24,25]. These symbiotic interactions are therefore considered to play an important role in protecting plants against abiotic stresses [26,27,28,29,30].

Inoculation with fungal endophytes helps increase the host plant’s tolerance to drought stress through various mechanisms [28,31,32,33]. Some endophytes can positively influence a plant’s biomass production by increasing the length and size of roots, which enhances its water and mineral nutrient uptake capacity, thus counteracting the negative effects of the stress [22,34,35]. Previous studies of gramineous species have found that plants inoculated with endophytic fungi develop higher levels of both aboveground and radical biomass, as well as higher leaf numbers, which improves drought tolerance as compared to non-inoculated plants [26,36,37,38]. In a wide range of plant species, endophytic fungi can boost gas exchange, chlorophyll content, photosynthetic rates, and the metabolism of substances involved in water stress responses [28,39,40,41]. Endophytic colonization can also regulate a plant’s osmotic capacity by modifying the concentrations of specific solutes, such as proline and certain sugars in the host tissue, which enhances the plant’s resistance to drought [42,43,44]. Several studies have found that, under drought stress conditions, endophytic fungi regulate the activation of antioxidant compounds and enzymatic activities involved in the protection against oxidative damage generated by drought [45,46,47]. This latter mechanism is considered crucial to alleviate the generation of ROS produced during drought in plant tissues [48,49]

In the Monte desert of La Rioja in Argentina, we observed that different strains of *Zopfiella erostrata* extensively colonized the roots of *Eragrostis cilianensis* (a desert grass), leading to the development of an abundant melanized mycelial network in the plant rhizosphere, which increases the water retention and nutrient mineralization of these plants [50,51]. In a previous study, we found that these strains play a key physiological role in the growth and survival of plants in these extreme environments [52]. However, further research is needed to understand the role of these fungal endophytes and their possible benefits under drought-stress conditions for crops of economic interest. In the present study, the horticultural crop plant tomato and wheat, an important cereal crop for human consumption, were selected. Both crops, which are primary sources of food worldwide, can grow in a broad range of environments. Thus, the objective of this study was to evaluate the effect of inoculation with *Z. erostrata* strains on stress mitigation among wheat and tomato plants grown under water deficit conditions. Overall growth variables, as well as physiological and biochemical parameters related to oxidative status, were determined in inoculated and non-inoculated plants subjected to 14 days of water deficit conditions.

## 2. Materials and Methods

### 2.1. Plant Experiment

The assay consisted of a randomized complete block design with three inoculation treatments for each plant species: (a) plants inoculated with the fungal endophyte *Zopfiella erostrata* 1; (b) plants inoculated with the endophyte *Zopfiella erostrata* 2; and (c) non-inoculated control plants. Two watering regimens were applied to the plants: (1) well-watered conditions (100% water-holding capacity) throughout the experiment and (2) drought conditions (water application was reduced by 60–50%) during the last two weeks before harvest. Each treatment was replicated ten times with respect to a total of 60 plants for each plant species.

### 2.2. Soil and Biological Materials

The growth substrate consisted of a mixture of loamy soil (soil composition is shown in Table 1) collected from IFAPA (Granada, Spain), which was diluted with quartz-sand (1:1, soil:sand, *v*/*v*), sieved (2 mm) and sterilized by steaming (100 °C for 1 h for three consecutive days).

Two fungal *Zopfiella aff. erostrata* endophytes (strains 1 and 2), which were isolated from arid environments in La Rioja, Argentina [51], were used in this study. These fungal strains related to coprophilous taxa of Lasiosphaeriaceae (Sordariales, Ascomycota) belong to the CRILAR fungal collection (accessions MK613228 and MK613213). The strains, which were maintained at 4 °C on malt-extract agar (MEA) plates, were periodically subcultured.

Wheat (*Triticum aestivum* L.) and tomato (*Solanum lycopersicum*, var. *muchamiel*) were used as experimental plant hosts. Seeds were superficially sterilized by washing and shaking with 10% NaClO for 15 min and four times with sterile distilled water. The seeds were then placed on sterile vermiculite at 25 °C to germinate. Two-week-old seedlings were transferred to plastic pots containing 1 kg of sterilized substrate (one seedling per pot).

### 2.3. Inoculation Treatments and Growth Conditions

As described by Reina et al. [53], endophyte inoculum was obtained from 500 mL Erlenmeyer flasks containing barley seeds previously hydrated and sterilized at 120 °C for 20 min. These flasks were inoculated with 3 cm^2^ agar plugs containing the active mycelium taken from the MEA cultures of each isolate. After two weeks of incubation at 25 °C, ten barley seeds homogeneously colonized with each fungal strain were added to the appropriate pots at sowing time just below the wheat/tomato seedlings.

The experiment was carried out under greenhouse conditions with natural and supplementary light at 25/15 °C and a photoperiod of 16 h. The plants were sown in pots containing 1 kg of sterile growth substrate. During five weeks, the plants were maintained under well-watered conditions (100% water holding capacity) to allow adequate fungal establishment and colonization of roots. After this growth stage, half of the pots were subjected to water stress for two weeks. At this time, the plants were allowed to dry until soil water content reached between 60–55% field capacity (3 d required), which corresponded to 10% volumetric soil moisture (determined experimentally in a previous assay). Plants were maintained under these conditions for an additional 12 d. To control the level of drought stress, the soil water content was measured daily using a ThetaProbe ML2 (Delta-T Devices Ltd, Cambridge, UK)) at the end of the afternoon and the amount of water lost was added to each pot to return soil water content to the desired 10% of volumetric soil moisture (60% of field capacity). However, during the 24 h period between each rewatering, the soil water content progressively decreased, reaching a minimum value of 50% of field capacity. Four replicates per treatment were performed for all evaluated parameters.

### 2.4. Parameters Measured

Photosynthetic efficiency and stomatal conductance were measured in vivo (prior to the plant’s harvest) 14 days after drought stress imposition. After seven weeks, the plants were harvested and the shoot and root systems were separated in 0.5 g aliquots which were stored at −80 °C for future determination of antioxidant enzyme activities (superoxide dismutase, catalase, ascorbate peroxidase and glutathione reductase) and other biochemical parameters such as proline, lipid peroxidation and hydrogen peroxide concentrations. The dry weight (DW) of shoots and roots was measured after drying in a forced hot-air oven at 70 °C for 48 h. The percentage of root colonization was estimated by visual observation of the fungal presence (melanized hyphae and mycelium) in the plant roots. Roots subsamples were stained following the procedure described by Barrow [54], and were observed under a binocular microscope to measure the colonization of mycelium according to the magnified intersections method [55]. Control plants were assessed to detect contaminations.

#### 2.4.1. Photosynthetic Efficiency (PSE) and Stomatal Conductance (gs)

The efficiency of photosystem II was measured using a FluorPen FP100 (Photon Systems Instruments, Brno, Czech Republic), which quantifies the yield of photosystem II as the ratio between the actual fluorescence yield in the light-adapted state (FV’) and the maximum fluorescence yield in the light-adapted state (FM’) [56]. This device facilitates a non-invasive assessment of plant photosynthetic performance by measuring chlorophyll fluorescence. FluorPen measurements were conducted on the second youngest leaf of six different plants of each treatment.

Stomatal conductance was measured two hours after the light turned on by using a porometer system (Porometer AP4, Delta-T Devices Ltd., Cambridge, UK) according to the user manual instructions. Stomatal conductance measurements were taken in the third youngest leaf from six different plants of each treatment.

#### 2.4.2. Antioxidant Enzymatic Activities

In order to determine enzymatic activities, 0.5 g of fresh leaves were ground in liquid nitrogen using a ceramic mortar until the tissue presented a homogeneous appearance. The material was then mixed with 1.5 mL of extraction buffer (50 mM Tris-HCl buffer pH 7.8, 1% albumin and 0.5% cysteine, pH 7.8). After homogenization, the extracts were filtered through nylon pads and centrifuged at 20,000× *g* for 20 min at 4 °C. Total protein was determined according to the Bradford method [57] with bovine serum as standard. The supernatant was used to measure the antioxidant enzymes at 25 °C using a spectrophotometer (Shimadzu UV-1603, Kyoto, Japan).

##### Superoxide Dismutase (SOD, EC 1.15.1.1)

SOD activity was measured according to the ferrocytochrome c method as described by MaCord and Fridovich [58]. The reaction mixture containing 0.1 mM EDTA, 1 mM xanthine, 1 mM citocrome c, 25 µL enzyme extract in 50 mM potassium phosphate buffer (pH 7.8) and sufficient xantine oxidase to produce a reduction in ferrocytochrome c at 550 nm. One unit of SOD activity was defined as the amount of enzyme required to inhibit the reduction in ferrocytochrome c by 50%, with the specific activity being expressed as U mg^−1^ protein.

##### Catalase (CAT EC 1.11.1.6)

CAT activity was determined by following the consumption of H_2_O_2_ (ꜫ = 23.5 mM^−1^ cm^−1^) at 240 nm for 1 min [59]. The reaction medium contained 0.1 M potassium phosphate buffer (pH 7.0), 18 mM H_2_O_2_ and 25 µg protein extract in a volume of 1 mL.

##### Ascorbate Peroxidase (APX EC 1.11.1.11)

APX activity was estimated by following the oxidation of ascorbate (ASC) at 290 nm (ꜫ = 2.75 mM^−1^ cm^−1^) for 1 min in 1 mL of the reaction medium containing 0.1 M Tris–acetate buffer (pH 6.4), 350 µM ascorbic acid, 170 µM H_2_O_2_ and 25 ug protein extract [60].

##### Glutathione Reductase (GR EC 1.6.4.2)

GR activity was assayed by following the oxidation of NADPH at 340 nm (ꜫ = 6.22 mM^−1^ cm^−1^) for 1 min [60]. The reaction medium contained 0.1 mM potassium phosphate buffer (pH 7.2), 1 mM glutathione oxidized (GSSG), 0.1 mM NADPH and 25 µg protein extract in a volume of 1 mL.

#### 2.4.3. Biochemical Analysis

##### Shoot Proline Content

Free proline was extracted from 0.5 g of fresh leaves [61]. The methanolic phase was used to quantify proline content. Proline was estimated by spectrophotometric analysis at 530 nm of the ninhydrin reaction according to the method described by Bates et al. [62].

##### Lipid Peroxidation

Lipid peroxidation was estimated by measuring malondialdehyde (MDA) content as determined by the thiobarbituric acid reaction according to the method described by Buege and Aust [63]. For the assay, 0.5 g of leaves was homogenized in 2 mL of 50 mM potassium phosphate buffer and centrifuged at 12,000× *g* for 30 min. The reaction medium contained 15% trichloroacetic acid, 0.375% 2-thiobarbituric acid, 0.01% butylated hydroxytoluene, 0.25 N chloride acid and 0.2 mL extract. The mixture was incubated for 30 min at 100 °C, cooled rapidly on ice, and centrifuged at 5500× *g* for 5 min. The supernatant was used to determine MDA at 535 nm using this compound as standard. Results were expressed as nm per mg^−1^ protein.

##### Hydrogen Peroxide Determination

Hydrogen peroxide (H_2_O_2_) content was measured according to the method described by Patterson et al. [64] with slight modifications. 0.5 g of the youngest fully developed leaves of each plant group were homogenized in a cold mortar with 5 mL 5% TCA containing 0.1 g activated charcoal and 0.1% PVPP. The homogenate was filtered and centrifuged at 18,000× *g* for 10 min. The supernatant was filtered through a millipore filter (0.45 mm) and used for the assay. A 200 mL aliquot was brought to 2 mL with 100 mM K-phosphate buffer (pH 8.4) and 1 mL colorimetric reagent was added. This reagent was made on the same day by mixing 1:1 (*v*/*v*) 0.6 mM potassium titanium oxalate and 0.6 mM 4–2 (2-pyridylazo) resorcinol (disodium salt). The samples were incubated at 45 °C for 60 min and the absorbance at 508 nm was recorded. The blanks were made by replacing leaf extract with 5% TCA.

### 2.5. Statistical Analysis

The data were analyzed by one-way ANOVAs using INFOSTAT software [65]. When significant differences were identified by ANOVA (*p* < 0.05), Duncan’s multiple range test for means comparison was applied [66]. Nonlinear principal component analysis (CATPCA) was used to determine the relation between the variables measured and the different treatments studied. We tested whether the growth variables of the plants (root dry weight, shoot dry weight, photosynthetic efficiency and stomatal conductance), antioxidant enzyme activities (SOD, CAT, APX and GR) and other biochemistry parameters (proline, MDA and hydrogen peroxide concentrations) differed among the six treatments evaluated (well-watered conditions, drought stress, fungal inoculation with *Z. erostrata* 1 y *Z. erostrata* 2).

## 3. Results

### 3.1. Shoot and Root Dry Weight

Under well-watered conditions, the shoot and root dry weight of wheat plants increased significantly following two treatments with the fungal endophytes as compared to control plants (Figure 1a). Under drought stress conditions, only the wheat plants inoculated with strain *Z. erostrata* 1 increased their shoot and root dry weight. The shoot and root dry weights of wheat plants subjected to drought stress decreased by half as compared to the well-watered plants. Root colonization by *Z. erostrata* 1 significantly increased root growth under both watering conditions, and *Z. erostrata* 2 considerably increased root growth, particularly under well-watered conditions (Figure 1c).

In inoculated tomato plants, both endophytes (*Z. erostrata* 1 and *Z. erostrata* 2) significantly enhanced shoot and root biomass regardless of the water regime applied (see Figure 1b,d, respectively). However, the largest increases in shoot and root biomass were observed in the plants inoculated with the strain 2 of *Z. erostrata* under both watering conditions.

### 3.2. Fungal Colonization

Fungal colonization of plants was not observed in the non-inoculated controls. In addition, no significant differences in the percentage of root colonization were detected between well-watered plants and those subjected to drought stress (see Figure 2a,b). The highest rate of fungal colonization was observed in plants inoculated with strain 1 of *Z. erostrata* under both watering conditions (≈65%), whereas the plants inoculated with strain 2 of *Z. erostrata* showed lower levels of colonization (≈35%). However, in tomato plants, the endophyte *Z. erostrata* 2 exhibited the highest root colonization levels, regardless of the watering conditions applied (see Figure 2a).

### 3.3. Photosynthetic Efficiency and Stomatal Conductance

The efficiency of photosystem II was evaluated by measuring chlorophyll fluorescence in plants 14 days after drought stress imposition. In general, the water regime did not alter the photosynthetic efficiency values for wheat plants. The inoculation with endophytes *Z. erostrata* 1 and *Z. erostrata* 2 did not change this parameter in the plants grown under well-watered conditions (Figure 3a). In contrast, under drought stress conditions, plants inoculated with *Z. erostrata* 1 exhibited higher photosynthetic efficiency than non-inoculated plants.

Under well-watered conditions, the stomatal conductance of wheat plants inoculated with both fungal strains was higher than that of non-inoculated plants (Figure 3c). Thus, plants inoculated with both types of endophytes achieved the highest stomatal conductance values (218 and 215, respectively). Likewise, under drought stress conditions, inoculation with both endophytes almost tripled this physiological parameter as compared to control wheat plants.

The photosynthetic efficiency of tomato plants was unaffected by fungal inoculation under well-watered conditions, as can be observed in Figure 3b. Conversely, under stressful conditions, the inoculated plants increased significantly photosynthetic efficiency as compared to the control plants. The stomatal conductance of tomato plants was significantly improved by fungal inoculation under both watering conditions (see Figure 3d). The improvement in this parameter was more evident in inoculated plants cultivated under well-watered conditions.

### 3.4. Antioxidant Enzymatic Activities

In order to determine the enzymatic activities involved in the antioxidant defense response triggered in wheat and tomato plants grown under well-watered and drought stress conditions, we analyzed SOD, CAT, APX, and GR activity.

The water regimen did not change SOD activity in wheat plants (Figure 4a). However, inoculation with both fungal endophytes decreased the SOD values considerably regardless of the water level applied. This decrease was more marked in wheat plants inoculated with fungus *Z. erostrata* 1 under the two watering conditions evaluated.

In tomato plants, drought stress induced an increase in SOD activity only in non-inoculated plants (Figure 4b). In contrast, the inoculation of tomato plants with *Z. erostrata* 1 and *Z. erostrata* 2 significantly decreased SOD activity under both well-watered and drought stress conditions, as compared to the corresponding control plants.

In wheat plants, CAT and APX activity decreased in all plants subjected to drought stress, whereas the enzymatic values were similar in both non-inoculated controls and inoculated plants (Figure 4c,d). The shoots of control plants cultivated under well-watered conditions exhibited the highest levels of CAT and APX activity, whereas plants inoculated with *Z. erostrata* 1 and *Z. erostrata* 2 exhibited a significant decrease in the activity of these enzymes. Tomato plants showed an almost opposite trend to wheat with respect to APX activity, as can be observed in Figure 4e. The activity of this enzyme was highest in control plants subjected to drought stress, whereas fungal inoculation resulted in the lowest APX values for the plants exposed to drought stress. Under the well-watered regime, fungal inoculation had an opposite effect on this activity. Thus, the APX levels of plants inoculated with *Z. erostrata* 1 and *Z. erostrata* 2 increased as compared to control plants. With regard to tomato plants, no CAT activity was detected in any of the plants evaluated.

Figure 4f shows that drought stress enhanced GR activity only in control wheat plants, whereas the inoculated plants exhibited a significant decrease in this enzymatic activity. This effect was more evident in the plants treated with *Z. erostrata* 2. In contrast, GR activity under well-watered conditions increased in the inoculated plants as compared to the control plants.

Conversely, as shown with regard to wheat, GR activity levels were low in all tomato plants subjected to drought stress, whose enzymatic values were unaffected by the endophytic inoculation. The control plants cultivated under well-watered conditions recorded the highest levels of GR activity, whereas inoculated tomato plants exhibited the lowest levels of this enzymatic activity (Figure 4g).

### 3.5. Biochemical Analysis

In wheat plants, the drought stress applied did not induce an accumulation of proline as compared to well-watered plants, except in the plants inoculated with fungus *Z. erostrata* 2, which almost tripled proline levels under these conditions (Figure 5a). In contrast, under well-watered conditions, plants inoculated with *Z. erostrata* 1 and *Z. erostrata* 2 showed a significant reduction in shoot proline content as compared to the control plants.

The water regime applied affected the proline content of the tomato plants, as can be observed in Figure 5b. The shoots of plants cultivated under drought conditions exhibited the highest levels of proline, as compared to the well-watered plants. Under well-watered conditions, plants inoculated with both endophytes showed a significant decrease in proline values with respect to control plants. However, under drought stress conditions, only those plants inoculated with endophyte *Z. erostrata* 2 exhibited significantly lower shoot proline content than non-inoculated tomato plants or those inoculated with *Z. erostrata* 1.

The amount of lipid peroxides formed increased in the wheat plants subjected to drought stress as compared to well-watered plants. However, the most remarkable result was observed in *Z. erostrata* 1-inoculated plants cultivated under drought conditions, whose lipid peroxide accumulation decreased significantly with respect to control plants exposed to drought conditions. When the wheat plants were subjected to well-watered conditions, fungal endophyte inoculation did not affect the levels of lipid peroxides accumulated, which were found to be similar in all plants (Figure 5c).

With regard to the lipid peroxidation of tomato plants, this parameter increased in all plants as a consequence of drought stress (Figure 5d). However, the increase was more pronounced in *Z. erostrata* 1-inoculated plants, which showed the highest values for this parameter. Under well-watered conditions, inoculation with *Z. erostrata* 2 considerably decreased lipid peroxidation levels, which, however, remained unchanged following treatment with the endophyte *Z. erostrata* 1.

Hydrogen peroxide levels varied depending on the fungal treatment used and the plant species. In wheat plants, drought stress decreased hydrogen peroxide concentrations both in plants inoculated with *Z. erostrata* 1 and in non-inoculated plants with respect to the well-watered plants (Figure 5e). Conversely, under well-watered conditions, hydrogen peroxide content decreased significantly in *Z. erostrata* 2-inoculated plants.

The shoots of tomato plants subjected to drought stress and inoculated with both endophytes showed the lowest levels of peroxide, whereas control plants exhibited the highest levels of this compound as compared to the well-watered plants (Figure 5f). However, under well-watered conditions, only inoculation with *Z. erostrata* 1 reduced hydrogen peroxide concentrations.

### 3.6. CATPCA Analysis

With regard to wheat, the dispersion plot shows the distribution of samples (dark blue diamonds) with respect to the parameters determined in the plants (Figure 6a). As can be observed, fungal inoculation with *Z. erostrata* 1 and *Z. erostrata* 2 under well-watered conditions (*Z. erostrata* 1_WW and *Z. erostrata* 2_WW) had a positive impact on the plant’s physical variables such as root and shoot dry weight, as well as photosynthetic efficiency and stomatal conductance, with respect to the non-inoculated controls (C_WW). In particular, inoculation with *Z. erostrata* 1_WW had a more pronounced effect on the growth parameters of wheat plants. Likewise, under drought conditions, inoculation with fungus *Z. erostrata* 1 produced a similar effect in the plants, as evidenced by the grouping of the samples in the same quadrant of the graph. Conversely, under drought conditions, inoculation with endophyte *Z. erostrata* 2 did not show any differences in SOD, MDA and proline levels with respect to control plants.

In tomato plants, fungal inoculations improved plant development under both water regimes evaluated (Figure 6b). Thus, the control treatments (C_WW and C_D) were grouped opposite from the physiological parameters measured. On the other hand, the fungi *Z. erostrata* 1_WW and *Z. erostrata* 2_WW enhanced GR and APX enzymatic activity under well-watered conditions, whereas, under drought conditions, these fungi modified mainly MDA and proline levels, as well as SOD activity.

## 4. Discussion

The adverse effects of drought stress on crop productivity have been widely reported [20,67]. One possible sustainable strategy to improve crop yield and tolerance to stress is through the incorporation of endophytic microbes [18]. The endophytic fungi evaluated in this study extensively colonize plant roots, leading to the development of a profuse melanized mycelium around the rhizosphere, which could be involved in improving water uptake and nutrient mineralization in the plants [51,52]. Although several endophytic fungi have been demonstrated to increase plant tolerance to water deficit conditions, the mechanisms involved in stress mitigation remain little understood [38]. Figure 7 shows a summary of the plant variables modified by endophytic fungi evaluated in this study.

In the present study, we evaluated the effect of the fungal endophytes, *Zopfiella erostrata* strain 1 and *Zopfiella erostrata* strain 2, on tomato and wheat plants grown under two different water regimes (well-watered conditions and drought stress). The results demonstrate that the growth responses of plants to *Zopfiella erostrata* inoculation were highly significant even under water deficit conditions. In general terms, both fungal endophytes improved the shoot and root development of the two plant species studied. However, the endophyte *Z. erostrata* 1 was found to perform better in terms of wheat plant growth, whereas fungus *Z. erostrata* 2 was found to have a more significant impact on both the shoot and root biomass of tomato plants regardless of the water conditions applied. These differences in growth responses between wheat and tomato plants have been detected in a diverse range of plant species [68]. This can be explained by the presence of specific mechanisms that regulate plant-fungus interactions [69]. Valli and Muthukumar [70] have demonstrated that endophytic inoculation with *Nectria haematococca* improved various growth parameters in tomato plants subjected to drought stress. These improvements in growth could be due to markedly higher nutrient and water uptake by the fungal hyphal network, as previously reported [28,39]. In addition, the results with respect to growth parameters correlated with the differential levels of fungal colonization, where the endophyte *Z. erostrata* 1 showed higher levels of colonization in the wheat plants, whereas *Z. erostrata* 2, to a great extent, colonized the tomato roots. Thus, the effects on growth are dependent upon the host involved, which is one of the factors that deeply influences this symbiotic relationship [71]. Therefore, it is vital to choose endophyte strains that are most beneficial to mitigate drought affecting a specific plant species [72]. In line with other studies, the extent of colonization in percentage terms in the roots evaluated was not altered by drought stress, suggesting that both strains are highly efficient in colonizing plants, even under the stress conditions studied [73,74]. However, the complex collaborative network regulating endophytic colonization under stress conditions, which is still little understood, requires further research. Indeed, given that the drought stress imposed in this study lasted only two weeks, the consequences of prolonged exposure to drought stress are not known.

Drought stress inhibits photosynthetic capacity and stomatal conductance (gs) in plants [1,75]. However, several studies have observed that endophytic fungi can counteract these adverse effects by making photosystem II and gas exchange more efficient [74,76]. In this study, the performance of photosystem II improved in both wheat and tomato plants subjected to stress when inoculated with the fungi. This could be related to the protective effect of photosynthetic capacity due to endophytic colonization [74,77]. Indeed, we found a positive correlation between increased stress resistance and photosynthetic efficiency, which even improved plant growth variables (Figure 6). The results obtained show that the endophytic colonization of host roots also had a marked effect on stomatal behavior, with inoculated plants exhibiting higher stomatal conductance values than non-inoculated plants under both water conditions. Previous research has demonstrated that endophytes can improve drought resistance by increasing photosynthetic and stomatal parameters under water deficit conditions [26,27]. Thus, our results suggest that endophytic colonization improved net photosynthetic rates both by increasing leaf gas exchange and by mitigating the drought effect on the photochemical capacity of PSII.

Plants have developed various antioxidative strategies to flush out reactive oxygen species (ROS) produced during drought stress [78]. Several studies suggest that increased stress tolerance is often associated with the enhancement of antioxidant defenses [79,80]. These defenses include antioxidant enzymes actively involved in plant ROS scavenging [31,81]. In general terms, our results showed that endophytic inoculation decreased SOD, CAT and APX activity in wheat plants regardless of the water regimen, whereas GR activity only decreased in inoculated plants subjected to drought. By contrast, previous studies have reported that plants inoculated with other endophytes isolated from desert plants increase SOD and CAT concentrations under water deficit conditions [38,39]. These contradictory results could be attributed to the dependence of plant antioxidant enzymes on the endophytic strains used and growth conditions, together with the genetic and environmental factors regulating these activities under different stress conditions [75,82]. Moreover, fungal inoculation may have contributed to the prevention of plant oxidative response through drought-avoidance mechanisms such as water supply to the plant. In line with our results, studies of strains of arbuscular mycorrhiza (AM) have also detected decreased antioxidant enzyme activity in plants subjected to drought stress [83,84]. Likewise, Pedranzani et al. [85] have also detected decreases in GR activity in perennial grass inoculated with *Rhizophagus irregularis* under abiotic stress conditions. With regard to enzymatic activity in tomato plants, SOD and APX levels were lower following fungal treatments under stress conditions as compared to non-stressed conditions. Similarly, other studies have found that inoculation with strains of endophytes and AM decrease antioxidant enzyme activity in plants under diverse abiotic stress conditions [86,87]. Previous investigations have reported that because fungal inoculation decreases ROS production or oxidative stress prevention, the host plant diminishes or maintains the level of antioxidant enzyme activity [87,88]. GR is an ascorbate–glutathione pathway enzyme, which reduces dehydroascorbate to ascorbate and is also indirectly involved in H_2_O_2_ removal [89]. In our study, we did not observe any changes in GR activity during drought, although H_2_O_2_ values decreased significantly in the inoculated tomato plants. In addition, no CAT activity was detected following any treatment, and APX levels were minimal in the tomato plants, which may indicate that hydrogen peroxide is scavenged by other non-enzymatic mechanisms such as antioxidant compound production [90,91]. In fact, numerous endophytic isolates have an antioxidant capacity through the production of phenols, sugars, and carbohydrates, which could enhance stress tolerance in host plants [92,93]. Another strategy to mitigate stress caused by endophytes could be melanized structures, which can also act as potential antioxidant agents that increase plant survival by counteracting the effect of free radicals produced during oxidative stress [31,39].

Fungal inoculation reduced hydrogen peroxide levels in all the plants subjected to stress, although the effect was more marked in tomato plants. This finding is in line with that observed in several AM-inoculated plants subjected to abiotic stresses such as drought [94,95], and salinity [96]. Porcel and Ruiz-Lozano [94] have reported that the reduction in H_2_O_2_ levels under drought stress conditions can be explained by enhanced CAT and APX enzyme activity. However, this correlation between the parameters mentioned above was not detected in our study. Although our results demonstrate the capacity of these endophytes to mitigate oxidative damage caused by drought, further research that provides a better understanding of the possible mechanisms involved is required.

Membrane lipid oxidation is a reliable measure of free-radical production and oxidative stress in plants [97]. In this study, lipid peroxides were quantified in the shoots of wheat and tomato subjected to water deficit. Under drought stress, the endophyte Z. erostrata 1 decreased the lipid peroxidation levels in wheat plants, as previously demonstrated with other endophytes [98]. Similarly, Zhu et al. [87] found that endophytic inoculation relieved membrane lipid peroxidation damage caused by metal stress in tomato seedlings. However, the endophyte *Z. erostrata* 1, which showed an opposite behavior to that of wheat in tomato plants, increased peroxidation levels as compared to non-inoculated plants exposed to drought. Given that the plants can respond in different ways to similar stress conditions and to fungal inoculation, the results are quite likely to vary depending on these factors.

Among the strategies more commonly used to cope with drought stress, many plants increase the osmotic potential of their cells through the accumulation of compatible osmolytes such as proline and mannitol, which are involved in osmotic adjustment [99,100]. The osmo-regulator proline scavenges free radicals and stabilizes altered redox potential due to drought [101,102]. In our study, proline accumulation levels varied considerably depending on the fungal strain and plants evaluated. In wheat plants subjected to drought, the two fungal strains evaluated behaved in completely opposite ways. Inoculation with endophyte *Z. erostrata* 1 decreased proline content, whereas *Z. erostrata* 2 significantly increased proline in these plants. Likewise, other studies have also reported a decrease in proline content in host plants in response to drought [103,104,105]. This decrease in the proline levels could be related to an improvement in all growth variables and to the high colonization levels of wheat plants inoculated with *Z. erostrata* 1. As these plants appear to be less affected by drought and have better water status than the non-inoculated plants, they do not need to accumulate proline. In tomato plants subjected to drought, *Z. erostrata* 2 decreased proline content, whereas *Z. erostrata* 1 maintained these values similar to those for control plants. This suggests that the high colonization levels of *Z. erostrata* 2 could increase water absorption by the roots of tomato plants and also improve growth parameters and stomatal conductance. Thus, the inoculated plants exhibit lower proline content due to enhanced water uptake by the roots [106,107]. Studies of osmoregulation in endophytic symbiosis have, up to now, been limited and sometimes contradictory.

## 5. Conclusions

In the present study, the effects on water stress of two unknown and poorly studied fungal endophytes were evaluated. This is therefore the first report to address the physiological and biochemical aspects of drought tolerance in wheat and tomato plants inoculated with two strains of *Zopfiella erostrata*. In general, inoculated plants exhibited higher tolerance to drought than non-inoculated plants, as evidenced by their enhanced shoot and root biomass production, higher photosynthetic efficiency and stomatal conductance, as well as lower peroxide content under these conditions. However, the data demonstrate the differential effects of endophytic inoculation depending on the plant species evaluated, suggesting a certain degree of compatibility between endophytic strains and host plant. With regard to biochemical aspects, proline accumulation, oxidative damage to lipids, and the activity of the four antioxidant enzymes measured varied considerably depending on the endophyte considered and the host plant. Thus, future research should focus on conclusively elucidating the mechanisms by which each of these endophytic strains improve plant drought resistance according to the host plant involved.

## Figures and Tables

**Figure 1 jof-09-00384-f001:**
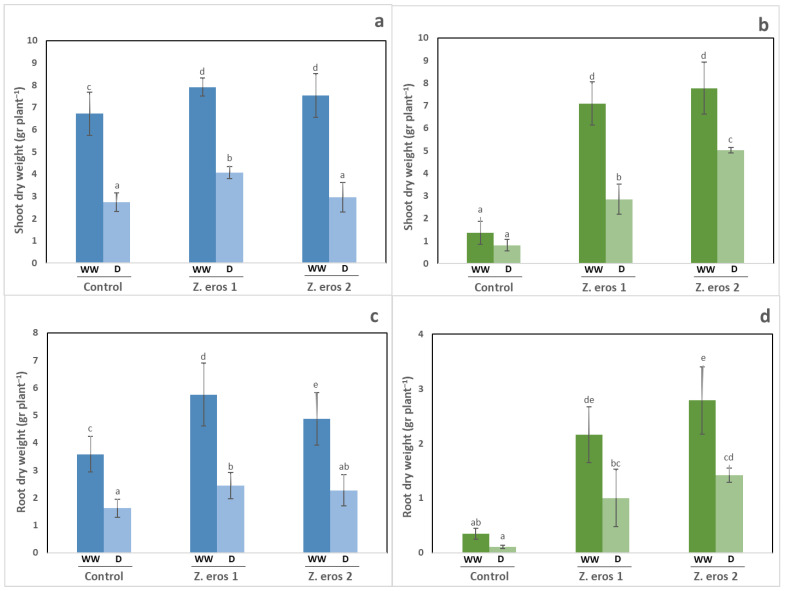
Shoot and root dry weight of wheat plants (**a**,**c**) and tomato plants (**b**,**d**) inoculated with the fungal endophytes *Z. erostrata* 1 y *Z. erostrata* 2 and cultivated under well-watered (WW) and drought stress (D) conditions. The data are the means ± standard deviation of ten replicates. Different letters indicate significant differences between all treatments, according to Duncan’s multiple-range test (*p* < 0.05).

**Figure 2 jof-09-00384-f002:**
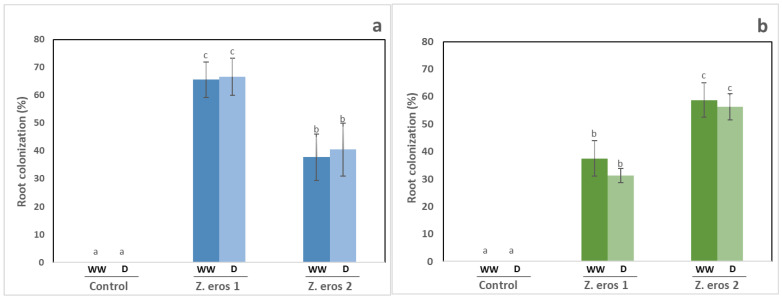
Percentage of root colonization in wheat plants (**a**), tomato plants (**b**) inoculated with the fungal endophytes *Z. erostrata* 1 y *Z. erostrata* 2 and cultivated under well-watered (WW) or drought stress (D) conditions. The data are the means ± standard deviation of ten replicates. According to Duncan’s multiple-range test (*p* < 0.05), different letters indicate significant differences between all treatments.

**Figure 3 jof-09-00384-f003:**
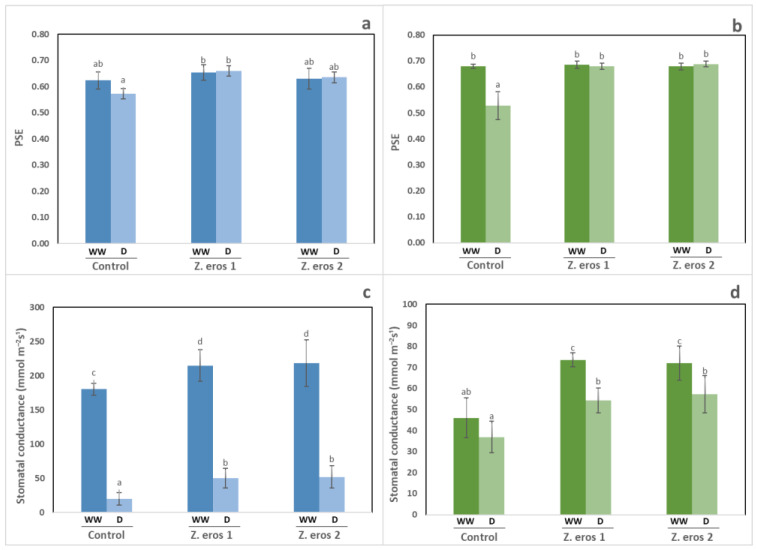
Photosynthetic efficiency (photosystem II) and stomatal conductance in wheat (**a**,**c**) and tomato (**b**,**d**) plants inoculated with the fungal endophytes *Z. erostrata* 1 y *Z. erostrata* 2 and cultivated under well-watered (WW) or drought stress (D) conditions. The data are the means ± standard deviation of four replicates. According to Duncan’s multiple-range test (*p* < 0.05), different letters indicate significant differences between all treatments.

**Figure 4 jof-09-00384-f004:**
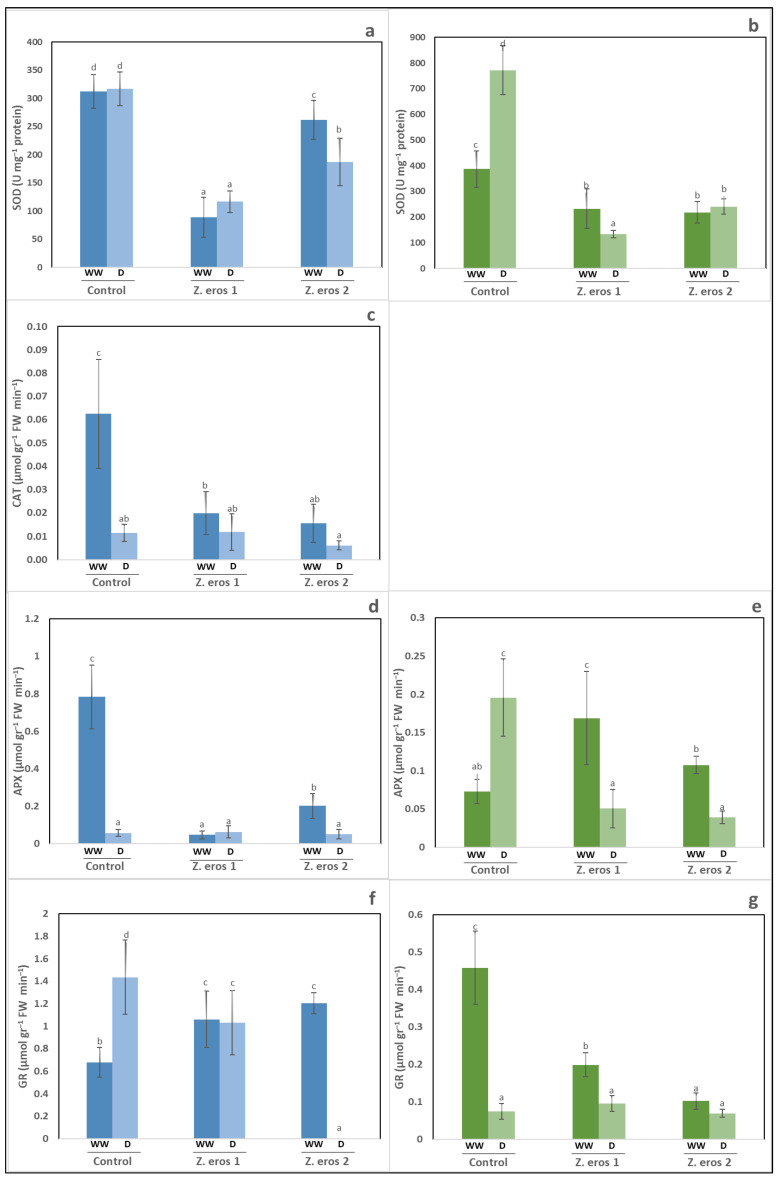
Activities of antioxidant enzymes (SOD, CAT, APX, GR) assayed in shoots of wheat (**a**,**c**,**d**,**f**) and tomato (**b**,**e**,**g**). Plants were inoculated with the fungal endophytes *Z. erostrata* 1 y *Z. erostrata* 2 under well-watered (WW) or to drought (D) conditions. Values represent the means ± standard deviation of four replicates. Significant differences are indicated by different letters above the bars according to Duncan’s multiple-range test (*p* < 0.05).

**Figure 5 jof-09-00384-f005:**
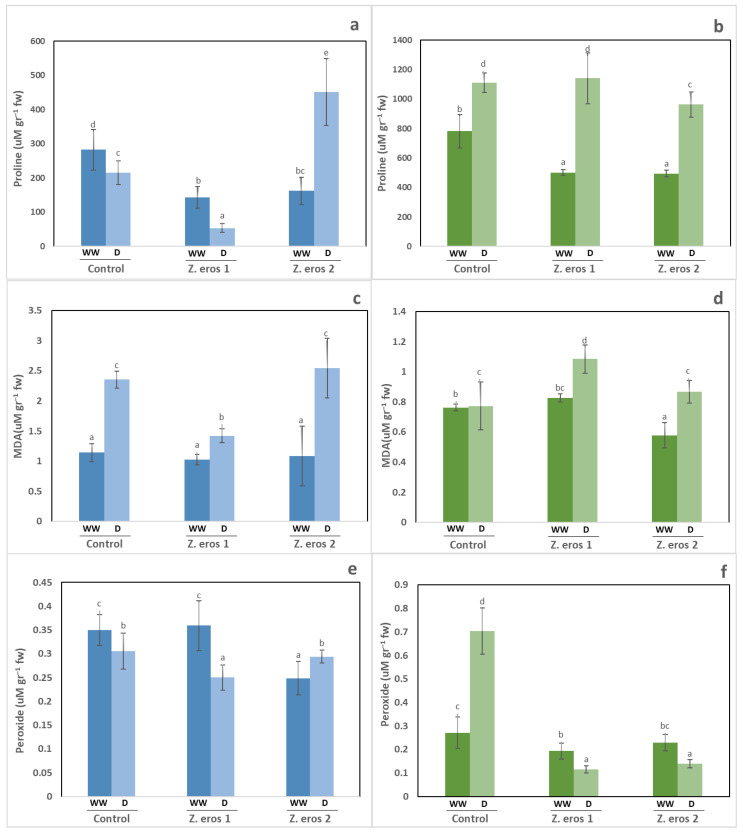
Proline content (**a**,**b**), MDA (**c**,**d**) and hydrogen peroxide content (**e**,**f)** in wheat and tomato plants respectively. Plants were inoculated with the fungal endophytes *Z. erostrata* 1 y *Z. erostrata* 2 and cultivated under well-watered (WW) or drought (D) conditions. Values are the means ± standard deviation of four replications. According to Duncan’s multiple-range test (*p* < 0.05), different letters indicate significant differences between all treatments.

**Figure 6 jof-09-00384-f006:**
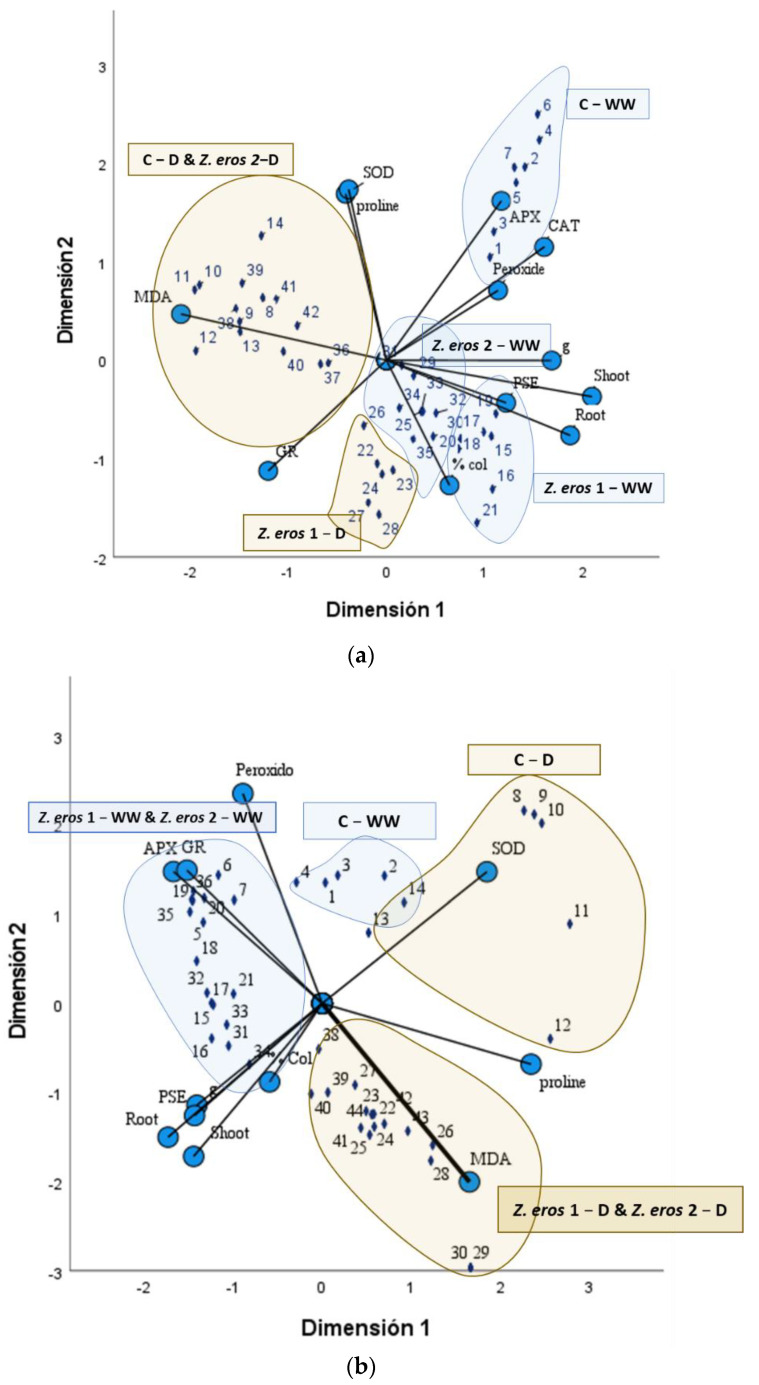
Categorical Principal Component Analysis (CATPCA) of wheat (**a**) and tomato (**b**) plant samples (indicated by dark blue diamonds) measured through antioxidant enzyme activities (SOD, APX, CAT, GR), biochemical analysis (MDA, proline, peroxide), Photosynthetic efficiency (PSE), stomatal conductance (g) and dry weight (shoot and root) (indicated by circle) in two different soil moisture conditions (well-watered conditions (WW) and subjected to drought (D)) and each fungus inoculated H1 or H3. The blue shading shows the distribution of the WW samples and the yellow shading shows the D.

**Figure 7 jof-09-00384-f007:**
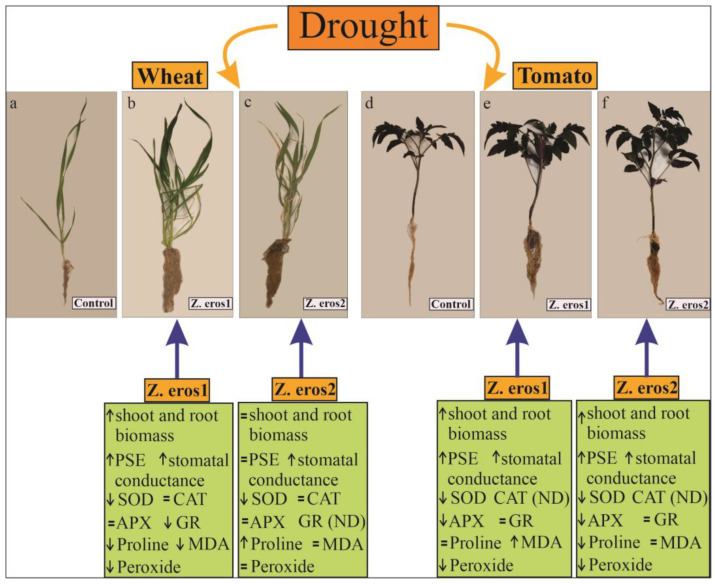
Effects triggered by two endophytic fungal strains (*Z. erostrata* 1 and *Z. erostrata* 2) to alleviate drought stress in wheat and tomato plants. Wheat plants (**a**–**c**) and tomato plants (**d**–**f**), uninoculated controls and plants inoculated with the fungal endophytes *Z. erostrata* 1 y *Z. erostrata* 2 subjected to drought stress conditions. ↑ arrows show an increase, while ↓ arrows show a decrease in the variable. The sign = indicates no significant changes in the parameter, and ND: indicates not detected.

**Table 1 jof-09-00384-t001:** Physicochemical characterization of the soil.

Physicochemical Parameters	Values
pH	8.32
CE (mS/cm)	0.367
% N total	0.0902
% C total	3.85
% C organic	0.74
% CaCO3	24.02
P (mg/kg)	10.13
Pb	60.59
As	12.41
Zn	74.46
Cu	23.57
Ni	47.41
Fe	22,875.49
Mn	394.88
Cr	81.01
Ca	60,209.38
K	13,942.2
S	1448.55
Sb	<LOD

## Data Availability

The datasets analyzed in this study are available from the corresponding author on reasonable request.

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
