# Peer review of "Fungal Endophytes Enhance Wheat and Tomato Drought Tolerance in Terms of Plant Growth and Biochemical Parameters"

_jof, 2023, doi:10.3390/jof9030384_

Round 1
Reviewer 1 Report
This manuscript reports interesting findings on the role of fungal endophytes isolated from a desert area in alleviating drought stress in crop plants.
The abstract of this paper is well-written and provides a clear and concise summary of the research. It includes the research question, methods used, key findings, and implications of the study. The abstract effectively captures the reader's attention and provides a good overview of the paper.
The introduction of this paper is strong and provides a clear problem statement. The authors clearly outline the issue they are investigating, which is the impact of drought stress on crops and developing a sustainable strategy to mitigate it. The introduction is written concisely and effectively sets the stage for the rest of the paper.
The result section is clearly written and the discussion provides a thorough analysis of the results. However, I recommend including representative images of the treated plants compared to the control to complete the well-written manuscript. The author claims extensive fungal colonization in the roots and around the rhizosphere and enhanced shoot and root biomass. Therefore, the figures must clearly show the difference between treated (with Z. eros 1 and 2) and untreated tomato and wheat plants, as described in the text and Figure 2.
Author Response
We agree with this comment. We added a figure 7 with images where the effects of both fungal strains on wheat and tomato plants subjected to drought conditions can be seen. In this figure we added also a network for adjusting to the requirement of another reviewer.

Reviewer 2 Report
1. Why no CAT activity was detected in the tomato?
2. In shoots of wheat treated with Z. erostrata 2, why no GR activity was detected?
3. Please draw a regulatory network Figure 7 response inoculated plants exhibited higher tolerance to drought than non-inoculated plants.
4. Other questions have been highlighted in the paper.

Author Response
- Why no CAT activity was detected in the tomato? RE: One possibility is that the CAT activity level in tomato plants was minimal, so the method used in this study cannot detect it. CAT is an antioxidant enzyme responsible for the degradation of H2O2 in a reaction where peroxide acts as a hydrogen donor and acceptor. In this case, the nonenzymatic antioxidant mechanisms could act through low molecular weight molecules (such as sugar, ascorbate, or glutathione) that remove the H2O2 and have the capability to enhance plant physiological status to effectively tolerate stress. However, additional research for a better understanding of the possible mechanisms involved is needed.
- In shoots of wheat treated with Z. erostrata 2, why no GR activity was detected? RE: We thought that the level of GR activity was basal in wheat plants inoculated with erostrata 2, so it was not detected. It is evident that the GR enzyme is not acting in ascorbate production and in H2O2 removal in stressed wheat plants. In addition, the process that regulates the responses to drought stress is complex and dynamic, involving different kinds of mechanisms depending on the plant species and fungal strain. Another possible proposed mechanism is melanized structures of endophytes, which can act as potential antioxidant compounds that increase plant survival in drought conditions.
- Please draw a regulatory network Figure 7 response inoculated plants exhibited higher tolerance to drought than non-inoculated plants. Re: We agree with this comment. We added figure 7 a regulatory network where the effects of both fungal strains on wheat and tomato plants subjected to drought conditions can be seen. We also including in this network images where the effects of both fungal strains on wheat and tomato plants subjected to drought conditions can be seen for adjusting to the requirement of another reviewer. Please check page 17 (Figure 7) in the new version of the manuscript.
- Other questions have been highlighted in the paper. RE: We accept this comment and modified the text highlighted as indicated by the reviewer.
Round 2
Reviewer 1 Report
Dear authors,
Figure 7 shows the obvious difference in terms of the root of biomass between treated and untreated plants. However, the figure did not show evidence of fungal colonization in the roots to support the data presented in Figure 2.
Moreover, the % of root colonization is shown in Figure 2. How did the authors calculate the % of root colonization? I just realized that the information is absent in the method section. Please furnish the information.
Finally, I don't think including the image (Figure 7) in the conclusion section is appropriate. The authors might want to consider adding the image to Figure 1 (in the same figure panel).
Author Response
Reviewer 1
Figure 7 shows the obvious difference in terms of the root of biomass between treated and untreated plants. However, the figure did not show evidence of fungal colonization in the roots to support the data presented in Figure 2. RE: Regarding the fungal colonization of roots, we did not find adequate images to include in the new version of the manuscript. However, the colonization of roots by these strains is well described in the previous article. Please check Miranda et al. 2020, to have a better visualization of this aspect.
Moreover, the % of root colonization is shown in Figure 2. How did the authors calculate the % of root colonization? I just realized that the information is absent in the method section. Please furnish the information. RE: Agreed. In the revised version of the manuscript we included more information about the measuring of the root colonization percent in wheat and tomato plants. Please check page 17 in the new version of the manuscript.
Finally, I don't think including the image (Figure 7) in the conclusion section is appropriate. The authors might want to consider adding the image to Figure 1 (in the same figure panel). RE: We agree with the reviewer on this point. However, we decided to move Figure 7 to the discussion section as a summary of the parameters affected by endophytic fungi in stressed plants for a better understanding of the readers. Please see page 13 in the revised manuscript.